



# Global, regional and seasonal analysis of total ozone trends derived from the 1995–2020 GTO-ECV climate data record

Melanie Coldewey-Egbers[1], Diego G. Loyola[1], Christophe Lerot[2], and Michel van Roozendael[2]

[1]Deutsches Zentrum für Luft- und Raumfahrt (DLR), Institut für Methodik der Fernerkundung, Oberpfaffenhofen, Germany
[2]Royal Belgian Institute for Space Aeronomy (BIRA-IASB), Brussels, Belgium

**Correspondence:** Melanie Coldewey-Egbers (Melanie.Coldewey-Egbers@dlr.de)

**Abstract.** In this study we present an updated perspective on near-global total ozone trends for the period 1995-2020. We use the GOME-type Total Ozone Essential Climate Variable (GTO-ECV) satellite data record which has been extended and generated as part of the European Space Agency's Climate Change Initiative (ESA-CCI) and European Union Copernicus Climate Change Service (EU-C3S) ozone projects. The focus of our work is the analysis of regional patterns in the ozone trend

as well as the investigation of its seasonal dependence. In the Southern Hemisphere we found regions that indicate statistically significant positive trends increasing from $0.6\pm0.5\%$ decade$^{-1}$ in the subtropics to $1.0\pm0.9\%$ decade$^{-1}$ in the middle latitudes and $2.8\pm2.6\%$ decade$^{-1}$ in the latitude band 60°–70°S. In the middle latitudes of the Northern Hemisphere the trend exhibits distinct regional patterns, i.e. latitudinal and longitudinal structures. Significant positive trends ($\sim1.5\pm1.0\%$ decade$^{-1}$) over the North Atlantic region as well as negative trends (-$1.0\pm1.0\%$ decade$^{-1}$) over Eastern Europe were found. Moreover, these

trends indicate a correlation with long-term changes in tropopause pressure. Total ozone trends in the tropics are not significant. Regarding the seasonal dependence of the trends we found only very small variations over the course of the year. However, we identified different behavior depending on latitude. In the latitude band 40°–70°N the positive trend maximizes in boreal winter from December to February. In the middle latitudes of the Southern Hemisphere (35°–50°S) the trend is maximum from March to May. Further south toward the high latitudes (55°–70°S) the trend denotes a relatively strong seasonal cycle which

varies from 2% decade$^{-1}$ in December and January to 3.8% decade$^{-1}$ in June and July.

## 1 Introduction

Monitoring the long-term evolution of ozone as a key constituent of the Earth's atmosphere is essential in order to assess the effectiveness of the Montreal Protocol (United Nations Environment Programme, 1986), which has been implemented to control and substantially reduce the amount of long-lived Ozone Depleting Substances (ODSs). The measures taken under the

Montreal Protocol and its amendments have led to a significant reduction of the total abundance of ODSs for the past two decades (Braesicke et al., 2018). Although the strong decrease in ozone amounts could be stopped, the expected positive trend is still not yet statistically significant in the near-global (60° N–60° S) total column amounts (Braesicke et al., 2018). In middle and high latitudes, the awaited ODS-related increase is masked by the large dynamically-induced interannual variability in ozone. Moreover, complex feedback mechanisms with climate change can lead to additional long-term variations in ozone



amounts, e.g., due to changing temperatures. Generally, the overall total ozone levels still remain lower than the pre-1980s values.

When considering height-resolved trends recent studies demonstrated that ozone in the upper stratosphere has started to recover (Steinbrecht et al., 2017; Braesicke et al., 2018; Arosio et al., 2019; SPARC/IO3C/GAW, 2019; Szeląg et al., 2020), whereas the evolution in the lower stratosphere does not provide evidence for a clear increase (Ball et al., 2018, 2019; Chipperfield et al., 2018; Szeląg et al., 2020). Some studies even show some signs of a decrease (though not statistically significant) in that altitude region (Ball et al., 2018; Braesicke et al., 2018). The increase in the upper stratosphere is seen in the middle latitudes of the Northern and Southern Hemisphere and in the tropics, and it has been attributed to both the decline of ODSs as well as the cooling induced by increasing amounts of greenhouse gases (Braesicke et al., 2018). Statistical confidence is largest in the Northern Hemisphere. Another confounding factor contributing to the overall trend in the total column is the evolution in the troposphere in particular in the tropics. However, changes in the tropospheric amount do not indicate a clear global pattern (Gaudel et al., 2018).

The aforementioned results reveal that the generation of consistent, multi-decadal time series of ozone with global coverage is key to assess long-term changes in ozone amounts with high reliability. As part of the European Space Agency's Climate Change Initiative (ESA-CCI) ozone project (Ozone_CCI) various Climate Data Records (CDRs) have been released that are based on satellite nadir-viewing sensors of the GOME-type (Global Ozone Monitoring Experiment) as well as limb- and occultation-type instruments. They allow for a comprehensive analysis of total ozone columns and vertical profiles at different spatial and temporal scales. The data records are freely available to users and, moreover, extended on a regular basis as part of the European Union Copernicus Climate Change Service (EU C3S) ozone project. In addition to ESA-CCI other projects and initiatives aimed at generating long-term data sets of total columns and profiles, that have recently been used for various trend analyses focusing on different questions (see, e.g., Weber et al., 2018; SPARC/IO3C/GAW, 2019; Szeląg et al., 2020, for more details).

Coldewey-Egbers et al. (2014) provided a global assessment of latitudinally and longitudinally resolved total ozone trends 1995–2013 using the first version of the GOME-type Total Ozone Essential Climate Variable (GTO-ECV) data record (Loyola et al., 2009; Loyola and Coldewey-Egbers, 2012; Coldewey-Egbers et al., 2015). The good spatial and temporal coverage of the underlying nadir-viewing satellite sensors allows the investigation of trends on regional scales instead of analyzing zonal averages only. The main findings of Coldewey-Egbers et al. (2014) were positive, though non-significant, ozone trends for large parts of the globe (60° N–60° S), in particular in the middle latitudes. Changes in the tropics were found to correlate significantly with the El Niño – Southern Oscillation (ENSO) variability. Longitudinal structures were found in both hemispheres. In the Northern Hemisphere enhanced values were found especially over the North Atlantic region and Scandinavia. It was concluded that strong natural variability, that is most pronounced in middle latitudes, hampers the expected ODSs-related onset of ozone recovery of being confirmed on a global scale, and that additional years of observations are required to thoroughly detect the awaited increase. Meanwhile the GTO-ECV data record has been expanded in time by 7.5 years (until December 2020) and, moreover, extended with three new additional sensors (see Sec. 2), which provides the opportunity to update the study.



**Table 1.** Overview of satellite sensors incorporated in GTO-ECV

| Sensor | Platform | Time period in GTO-ECV |
|---|---|---|
| Global Ozone Monitoring Experiment (GOME) | European Remote Sensing satellite 2 (ERS-2) | 07/1995 – 12/2004 |
| Scanning Imaging Absorption Spectrometer for Atmospheric Chartography (SCIAMACHY) | Environmental Satellite (ENVISAT) | 08/2002 – 12/2004 |
| Ozone Monitoring Instrument (OMI) | Aura | 10/2004 – 12/2020 |
| Global Ozone Monitoring Experiment - 2 (GOME-2) | Meteorological Operational satellite A (MetOp-A) | 01/2007 – 12/2017 |
| Global Ozone Monitoring Experiment - 2 (GOME-2) | Meteorological Operational satellite B (MetOp-B) | 01/2013 – 12/2020 |
| Tropospheric Monitoring Instrument (TROPOMI) | Sentinel-5 Precursor satellite (S-5P) | 05/2018 – 12/2020 |

During the past years, the investigation of the longitudinal structure in ozone changes and the evaluation of their seasonal dependence came into focus. Arosio et al. (2019) and Sofieva et al. (2021) have shown that height-resolved trends exhibit a considerable dependence on longitude. For the temperature in the stratosphere, which is key indicator of climate change, a seasonal variation in the trends was already detected (Funatsu et al., 2016; Khaykin et al., 2017), and, since temperature changes are closely coupled to changes in ozone (and greenhouse gases), correlations in trend patterns can be anticipated. In

a recent study by Szeląg et al. (2020) a positive inter-relation in the lower stratosphere and a negative correlation in the upper stratosphere was found.

     This study focuses on the evaluation of total ozone trends using the updated and extended GTO-ECV data record. Latitudinal and longitudinal structures are analyzed, and signs for a seasonal dependence in total ozone changes are investigated. Section 2 contains a brief description of the GTO-ECV data record including its recent updates and extensions, and Sect. 3 provides the

details of the regression model that is applied for estimating the trends. Results related to spatial patterns in the annual mean trend are discussed in Sect. 4.1 and the seasonal variation is presented in Sect. 4.2. A summary (Sect. 5) completes the paper.

## 2    The GOME-type Total Ozone Essential Climate Variable data record

GTO-ECV is a merged climate data record of total ozone columns which incorporates measurements from a series of nadir-viewing satellite instruments mounted on low Earth orbit platforms. By now, it covers the past two and a half decades starting in

July 1995. The latest version of GTO-ECV has been generated in the framework of the ESA-CCI+ ozone project. Here we will provide a brief summary of the data record, and explain the merging approach and recent updates. For detailed descriptions of the predecessor versions we refer to Loyola et al. (2009), Loyola and Coldewey-Egbers (2012), Coldewey-Egbers et al. (2015, 2020), or Garane et al. (2018).

     An overview of all satellite sensors included in GTO-ECV is given in Table 1. The third column denotes the time period

for which data from the respective sensor are ingested in GTO-ECV. We use total ozone columns retrieved with the GOME Direct Fitting version 4 (GODFIT_V4) algorithm (Lerot et al., 2014; Garane et al., 2018), that is applied to all sensors. A full





reprocessing of the entire time series of GOME, SCIAMACHY, OMI, and GOME-2A/-2B has been realized in the framework of the ESA-CCI and the EU/ECMWF C3S ozone projects. Moreover, GODFIT is the baseline retrieval algorithm for the offline total ozone column products of TROPOMI/S-5P, in addition to the GOME Data Processor (GDP) algorithm, which is used for
the corresponding near real-time products (Inness et al., 2019; Spurr et al., 2021).

Geophysical validation of the GODFIT_V4 level-2 total ozone column products from the individual nadir sensors with respect to ground-based reference instruments, such as Brewer, Dobson, and zenith-sky spectrometers, evidences a very good quality (Garane et al., 2018, 2019). Owing to the common retrieval algorithm the inter-sensor consistency of the selected instruments is overall extremely high. This is an excellent prerequisite for homogenizing and merging of the individual time
series, which is performed on daily gridded level-3 products that are generated from the corresponding level-2 products. In order to account for possible remaining biases and/or drifts between the individual data sets, adjustments are applied to GOME, SCIAMACHY, GOME-2A, GOME-2B, and TROPOMI, which all benefit from long overlap periods with OMI, whose measurements are used as a reference basis. The adjustments depend on latitude and time. Compared to the latest predecessor version of GTO-ECV, as described in Garane et al. (2018), TROPOMI has now been included as an additional sensor, and minor
changes related to the time periods during which the individual sensors are used in GTO-ECV have been made (see Table 1 for the current time periods). The validation of GTO-ECV total ozone columns against ground-based observations reveals a very good overall agreement (i.e., similar to the validation of the level-2 data) and an excellent long-term stability. Especially the latter makes it useful for addressing open questions related to the evolution of the ozone layer, in particular to ozone recovery and inter-annual variability (Coldewey-Egbers et al., 2014; Chipperfield et al., 2018; Weber et al., 2018; Eleftheratos
et al., 2019), or the evaluation of extreme values (Dameris et al., 2021). Moreover, various other total ozone data sets such as chemistry-climate model simulations, re-analysis data, or similar satellite-based records could benefit from a comparison with GTO-ECV in order either to assess the capability of the models' systems or to identify drawbacks, e.g., jumps or drifts in one of the satellite-based records (Loyola et al., 2009; Chiou et al., 2014; Coldewey-Egbers et al., 2020).

The GTO-ECV data record is freely available from the Copernicus Climate Data Store (CDS; https://cds.climate.copernicus.
eu). It provides monthly mean total ozone columns as well as the corresponding standard deviations and standard errors from July 1995 through December 2020 on a spatial grid of $1° \times 1°$ in latitude and longitude in a user-friendly NetCDF format following the climate and forecast metadata conventions. Along with other long-term total ozone data records, GTO-ECV is used in relevant international assessments and reports such as the World Meteorological Organization (WMO) / United Nations Environment Programme (UNEP) Scientific Assessment of Ozone Depletion (e.g., Braesicke et al., 2018), the State
of the Climate published as a supplement to the Bulletin of the American Meteorological Society (BAMS, e.g., Weber et al., 2021b) and the Working Group I contribution to the Sixth Assessment Report of the Intergovernmental Panel on Climate Change (IPCC, 2021).



## 3 Multivariate linear regression model

In this section, a brief description of the regression model, that is used for trend estimation, and a short summary of the selected
explanatory variables are provided. We derive total ozone trends using a standard multivariate linear regression (MLR) model
which is applied to deseasonalized monthly mean ozone anomalies obtained from the GTO-ECV data record. Please note that
for trend analyses we converted the original product to a slightly larger grid size of $5° \times 5°$. The MLR is used in the form given
by

$$\Delta O3(m) = a + b \cdot m + c \cdot \mathrm{QBO30}(m) + d \cdot \mathrm{QBO50}(m) + e \cdot \mathrm{SF}(m) + f \cdot \mathrm{MEI}(m) + g \cdot \mathrm{(A)AO}(m) + \mathrm{X}(m). \quad (1)$$

The equation quantifies the relationship between ozone anomalies and several explanatory variables describing natural forc-
ings. $\Delta O3(m)$ is the deseasonalized ozone anomaly (in %), $m$ is the number of months after the initial time, and $a - g$ are
the model fit coefficients calculated using a standard least squares algorithm. Seasonal variability can be accounted for by
expanding the respective coefficient as, for example, for the trend term (second term on the right hand side of Eq. 1)

$$b = b_0 + \sum_{k=1}^{N_b} b_{2k-1} \sin(2\pi k m/12) + b_{2k} \cos(2\pi k m/12), \quad (2)$$

where $b_0$ is the annual mean trend. Depending on the choice of $N_b$ it is possible to account for annual, semiannual, 4 months,
and, if necessary, shorter-term variations. We apply the MLR to the GTO-ECV product introduced in Sect. 2), but for trend
estimation we use a slightly shorter time period from January 1997 through December 2020. 1997 is regarded as an adequate
turnaround point for ozone after ODSs have peaked (Harris et al., 2008), and thus it should be an appropriate starting point for
the investigation of the recovery period of ozone. However, the optimum inflection point may vary with latitude and altitude
(SPARC/IO3C/GAW, 2019), and we have to keep in mind that trend estimates may be sensitive to starting and end point of the
regression (Harris et al., 2015).

Compared to the MLR used in Coldewey-Egbers et al. (2014) we added one more proxy that is the Arctic Oscillation (AO)
index for all GTO-ECV grid cells in the Northern Hemisphere and the Antarctic Oscillation (AAO) index for all grid cells in
the Southern Hemisphere, respectively, in order to account for large-scale circulation processes in the middle latitudes (Reinsel
et al., 2005; Steinbrecht et al., 2011; Weber et al., 2018). Moreover, we extended the latitude range for which we calculate the
trend to $70°S–70°N$. The sources and names of the selected explanatory variable time series are given in Table 2. All covariates
have been normalized to zero-mean and unit-variance, and the time series are shown in Fig. A1. A potential non-negligible
autocovariance of the residuals $X(m)$ is accounted for by applying a Cochrane-Orcutt transformation (Cochrane and Orcutt,
1949), and thus, the regression is performed in an iterative procedure. In general, a fit coefficient is considered statistically
significant when its absolute value is greater than 2 times its error (95% confidence interval).

In general, between 70 and 90% of the ozone variance are explained by the full MLR as described in Eq. 1, when it is applied
to total ozone columns (see also Coldewey-Egbers et al. (2014, their Fig. 2a ). In the middle latitudes this variance is dominated
to a large extend by the seasonal variability. In the tropics ozone variability is additionally induced by the QBO, the solar cycle,
and ENSO (Toihir et al., 2018; Bencherif et al., 2020). Toward the high latitudes of the southern hemisphere the magnitude





**Table 2.** Sources of explanatory variable time series used in the MLR (Eq. 1). Figure A1 shows the time series (normalized to zero-mean and unit-variance) for all individual covariates.

| Variable | Denotation (Source) |
|---|---|
| QBO30($m$) | Zonally averaged equatorial wind at 30 hPa |
| | (https://www.cpc.ncep.noaa.gov/data/indices/qbo.u30.index) |
| QBO50($m$) | Zonally averaged equatorial wind at 50 hPa |
| | (https://www.cpc.ncep.noaa.gov/data/indices/qbo.u50.index) |
| SF($m$) | 10.7 cm Solar Flux |
| | (ftp://ftp.seismo.nrcan.gc.ca/spaceweather/solar_flux/monthly_averages/solflux_monthly_average.txt) |
| MEI($m$) | Multivariate ENSO Index Version 2 |
| | (https://psl.noaa.gov/enso/mei/data/meiv2.data) |
| AO($m$) | Arctic Oscillation |
| | (https://www.cpc.ncep.noaa.gov/products/precip/CWlink/daily_ao_index/monthly.ao.index.b50.current.ascii.table) |
| AAO($m$) | Antarctic Oscillation |
| | (https://www.cpc.ncep.noaa.gov/products/precip/CWlink/daily_ao_index/aao/monthly.aao.index.b79.current.ascii.table) |

of the explained variance using our MLR decreases to about 50% suggesting that part of the forcing of ozone variability is missing in this region. In contrast to our model in other ozone trend and variability studies the poleward heat flux is included as additional and significant proxy (de Laat et al., 2015; Toro A. et al., 2017; Weber et al., 2018, 2021a), which might explain the lower values of the explained variance we achieve with the MLR.

The quasi-biennial oscillation (QBO) signal is approximated by both 30 hPa and 50 hPa equatorial zonally averaged monthly
winds denoted as QBO30($m$) and QBO50($m$) in Eq. 1. The QBO is the dominant source of stratospheric variability in the equatorial region (10° S–10° N), but has also an impact in the subtropics (with the opposite sign) and to some extend in the middle latitudes (e.g., Steinbrecht et al., 2006; Frossard et al., 2013). The mean period is about 26-28 months. The QBO MLR fit coefficients $c$ and $d$ are shown in Fig. 1 (a) and (b). During the QBO's westerly phase there is a positive correlation between wind anomalies at 30 hPa and ozone and a negative correlation between wind anomalies at 50 hPa and ozone in the inner
tropics. They are related to changes in up- and down-welling processes (Baldwin et al., 2001).

In Eq. 1 SF($m$) denotes the 10.7 cm radio flux and accounts for the impact of the 11-year solar cycle. Between solar activity and total ozone in the tropics, a positive correlation (though barely significant) exists which is shown in Fig. 1 (c). Enhanced ozone levels occur during solar maxima and vice versa. While the correlation pattern is basically zonally symmetric at low latitudes, it exhibits a longitudinal variance (though not statistically significant) at middle to high latitudes due to transport
and dynamics (Steinbrecht et al., 2003; Coldewey-Egbers et al., 2014). However, an accurate quantification of the impact is difficult since only slightly more than two solar cycles are covered by GTO-ECV.



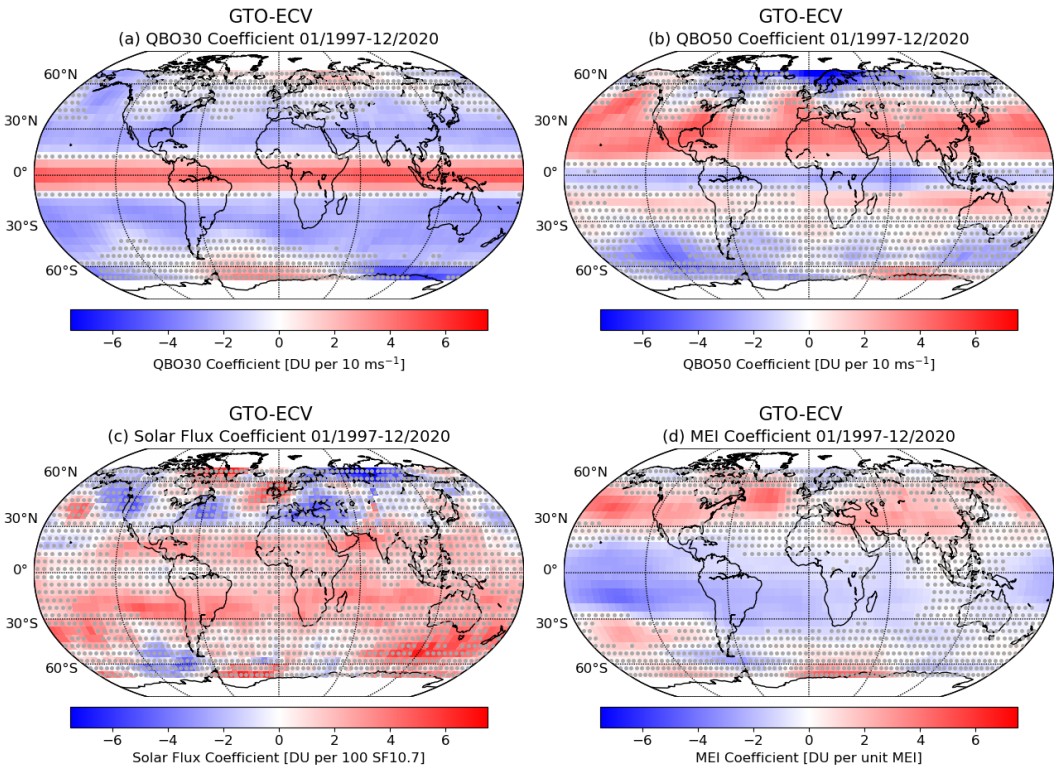

**Figure 1.** MLR fit coefficients for the explanatory variables (a) QBO at 30 hPa (QBO30), (b) QBO at 50 hPa (QBO50), (c) solar flux (SF), and (d) Multivariate ENSO Index (MEI). The results are expressed in DU per unit of the explanatory variable. Grey dots denote that the term is statistically not significant in this grid cell.

The correlation between ozone and the ENSO pattern, that is represented by the Multivariate ENSO Index (MEI) and denoted by $\mathrm{MEI}(m)$ in Eq. 1, is negative for almost the entire tropics and has its maximum over the Pacific (see Fig. 1 (d)). Significant changes in the convection pattern induced by El Niño or La Niña events lead to changes in tropospheric ozone amounts,
whereas the stratospheric columns remain more or less unchanged. Cold (La Niña) events lead to an increase in tropospheric (and thus total) ozone mainly over the eastern Pacific due to reduced convection. On the other hand, warm (El Niño) events and enhanced convection lead to a decrease in ozone, respectively (Steinbrecht et al., 2006).

The left panel of Fig. 2 shows the fit coefficient for the Arctic Oscillation (AO) index (applied in the Northern Hemisphere only) and the Antarctic Oscillation index (applied in the Southern Hemisphere only). In the Northern Hemisphere the correla-
tion is significant mainly over the eastern part of North America, the Atlantic Ocean, Europe, and over Eurasia. The observed pattern over the North Atlantic sector suggests a strong correlation with the North Atlantic Oscillation (NAO) pattern which is a part of the AO. During the positive phase of the AO (or the NAO) the pressure gradient between the Icelandic Low and the Azores High is stronger than normal, i.e. lower than normal pressure in the north and higher than normal pressure further



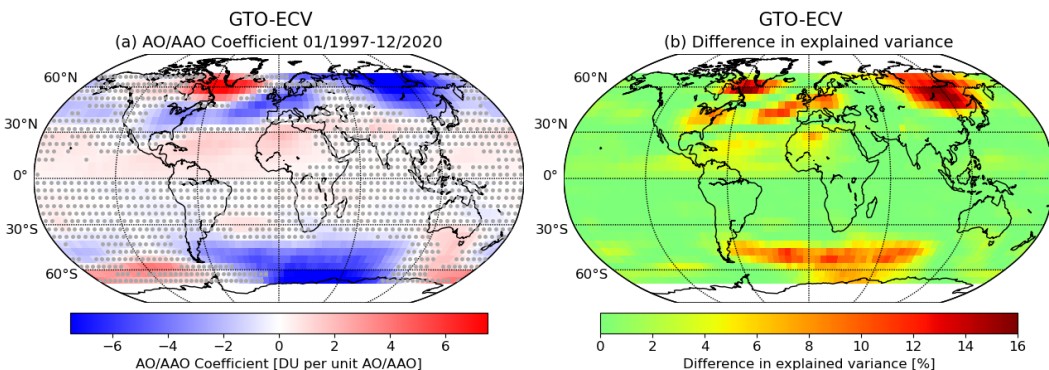

**Figure 2.** Left panel: MLR fit coefficient for the additional explanatory variables AO (applied in Northern Hemisphere) and AAO (applied in the Southern Hemisphere) in DU per unit AO/AAO index, respectively. Right panel: Difference in explained variance (in %) between the MLR including the AO/AAO proxy and the MLR without the additional proxy.

south. Over Iceland the corresponding lower tropopause altitude leads to an increase in total ozone (i.e. positive correlation between AO and ozone), whereas further south a higher than normal tropopause altitude leads to a decrease in total ozone, i.e. a negative correlation (cf., Appenzeller et al., 2000; Steinbrecht et al., 2011; Zhang et al., 2017). Over Eurasia, poleward of 40°N, the correlation is significantly negative, i.e. ozone is lower during the positive phase of the AO. The positive phase of the AO is related to a strengthened polar vortex, lower than normal air pressure, and enhanced ozone depletion over high latitudes (Liu and Hu, 2021). According to Thompson and Wallace (1998) or Zou et al. (2005) this correlation is more pronounced over Eurasia than over the North Atlantic region. Thompson and Wallace (2000, their Fig. 12) also stated that the area of a negative correlation between AO and ozone reaches from the polar cap to 40°N which is in good agreement with Fig. 2. The zonal asymmetry in the correlation indicating a positive maximum over the Labrador Sea is reproduced as well. The positive correlation between AO and ozone in the tropics is probably related to a lowering of the tropopause in conjunction with temperature changes (Thompson and Wallace, 2000). For the Southern Hemisphere and the corresponding AAO a similar relationship as for the AO was found. Ozone is reduced during the positive phase of the AAO from 60°W to 120°E. The zonal asymmetry found by Thompson and Wallace (2000) expressed by a positive correlation northwest of the Antarctic Peninsula is reproduced, but in the region south of Australia, we see a discrepancy between our results (slightly positive correlation) and their findings (negative correlation).

The right panel of Fig. 2 denotes the difference in the explained variance between the new MLR including the AO/AAO proxies and the old model without AO/AAO proxies. The explained variance is enhanced in those regions discussed above which show a significant correlation with total ozone. In the Northern Hemisphere a maximum increase of 15–17% is found in the 55°–60°N latitude belt around 60°W and poleward of 45°N from 90°E to 150°E. In the Southern Hemisphere we found an increase of ∼9% in the latitude range 45°–60°S from 60°W to 90°E. Note that we do not expect significant changes regarding

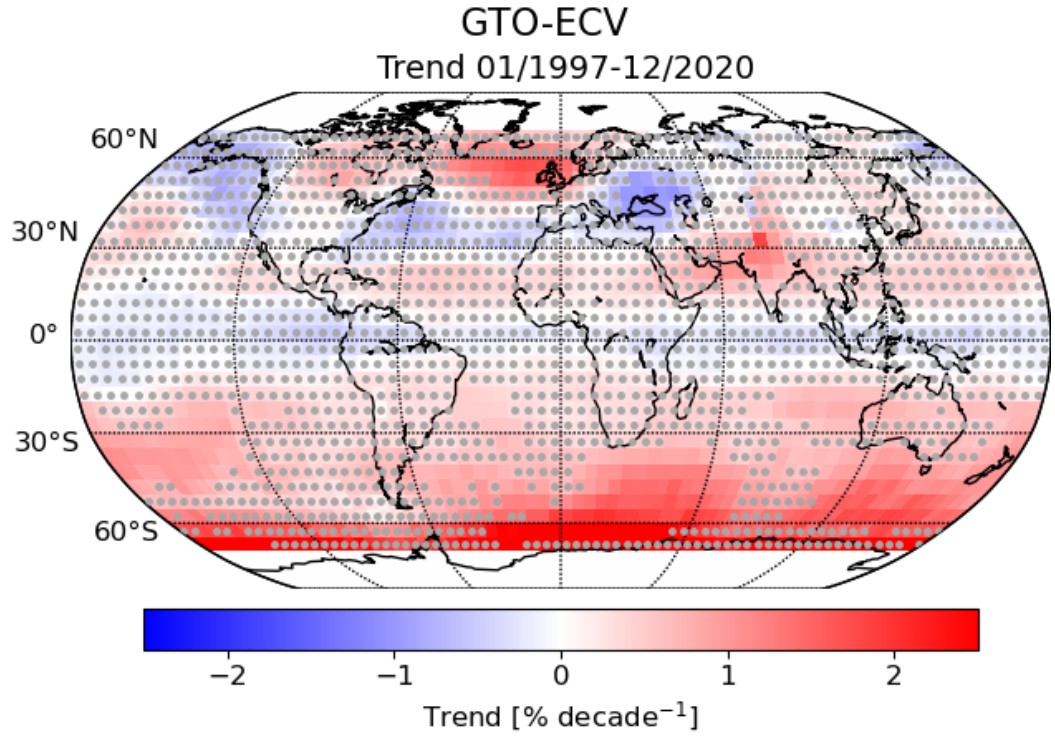

**Figure 3.** Total ozone trend 1997–2020 in % decade$^{-1}$ derived from the GTO-ECV data record. Grey dots denote that the trend is statistically not significant at the $2\sigma$ uncertainty level.

the ozone trend when we include the additional terms in the MLR, but uncertainty should be further reduced (see also Weber et al., 2018).

## 4 Results

### 4.1 Annual mean trends

The annual mean total ozone trend in percent per decade derived from the GTO-ECV data record for the time period 1997–2020 and the latitude range 70° S–70° N is shown in Fig. 3. The MLR (Eq. 1) has been applied to all $5° \times 5°$ grid cells separately, grey dots denote that the trend in a grid cell is statistically not significant at the $2\sigma$ uncertainty level.

In the middle latitudes of the Northern Hemisphere the trend indicates a longitudinal structure although the trend is statistically not significant for almost the entire latitude band. The zonal average of the trends is close to zero and not significant. Small regions which show significant trends are the North Atlantic between 50°–60° N as well as eastern Europe and the Black Sea. Over the northern Atlantic the trend is positive ($\sim 1.5 \pm 1.0\%$ decade$^{-1}$) and over eastern Europe and the Black Sea the trend





is negative ($\sim -1.0 \pm 1.0\%$ decade$^{-1}$). A similar pattern was found by Sofieva et al. (2021, their Fig. 10) and by Arosio et al. (2019). Their height-resolved trends indicate significant positive trends in the upper stratosphere above 40 km with a maximum over Scandinavia and the North Atlantic. Non-significant negative trends are found by Sofieva et al. (2021) at about 25 km in the same region as found here for the total column, i.e. in the northern Pacific, the Atlantic (30°–45°N), as well as over eastern Europe and the Black Sea. These longitudinal structures suggest that the trends seem to have non-negligible contributions

from dynamical and climate change effects (Zhang et al., 2018; Sofieva et al., 2021), which cannot be easily separated from the ODSs-related trends. The spatial trend pattern described above indicates some correlation with the fit coefficients for the AO/AAO proxies (Fig. 2). Since changes in ozone related to this forcing are at least partly induced by changes in tropopause altitude we investigate this relationship later in more detail. The significant positive trend we found north of India is mainly caused by artificial outliers in the GTO-ECV data record. This region is impacted by regular data gaps in the GOME/ERS-2 time series (see also https://atmos.eoc.dlr.de/app/calendar) and should be omitted for the analysis and interpretation of ozone

trends.

In the tropics the trend does not show a noticeable longitudinal variation. Moreover, it is not significant from 30°N to 20°S, which is in good agreement with Braesicke et al. (2018). It is assumed that the total ozone trend in this region is governed by various dynamical and chemical processes at different altitudes competing with each other. In particular, an increase in

ozone is expected in both the troposphere and the upper stratosphere. An increase in the troposphere is most probably due to changes in surface emissions of precursor constituents and the increase in the upper stratosphere is expected to be related to both decreasing amounts of ODSs and an increase in greenhouse gases. The latter leads to a cooling which will in turn lead to an increase of ozone levels (Meul et al., 2016; Keeble et al., 2017; Toihir et al., 2018). On the other hand, changes in the tropical lower stratosphere are governed by transport processes. A negative trend in ozone due to an enhanced Brewer-Dobson

circulation (BDC) as a consequence of increasing amounts of greenhouse gases is expected and has already been detected (Eyring et al., 2010; Ball et al., 2018, 2019; Szeląg et al., 2020).

Southward of 20° S some regions indicate significant positive trends, mainly in the Pacific, south of Africa, and around Australia. Almost 45% of all grid cells between 30°S and 70°S denote significant trends. The positive trend increases from $0.6 \pm 0.5\%$ decade$^{-1}$ in the southern subtropics to $1.0 \pm 0.9\%$ decade$^{-1}$ in the middle latitudes and $2.8 \pm 2.6\%$ decade$^{-1}$ in the

latitude band 60°–70°S. The latitudinal structure of the estimated trends is quite similar to the results described in Weber et al. (2018), who analyzed annual mean zonal mean data from different data sets for the period 1979–2016. In their paper only in the middle latitudes of the Southern Hemisphere significant positive trends in zonal mean ozone of about 0.5–1.0% decade$^{-1}$ were detected, whereas they were insignificant in most parts of the tropics and Northern Hemisphere middle latitudes. For the latter regions adding four more years of measurements did not change the conclusions; the zonal mean trend is still near zero

and insignificant. In particular, our analysis does not confirm the significant positive trend found by Weber et al. (2018) in the northern subtropics (20°–30°N).

When we compare our new results derived for the annual mean ozone trend using the extended (by 7.5 years and additional satellite sensors) GTO-ECV data record (Fig. 3) with the findings from Coldewey-Egbers et al. (2014, their Fig. 1(a)), we notice that the overall spatial pattern changed, in particular in the tropical region and in the Southern Hemisphere. The significant





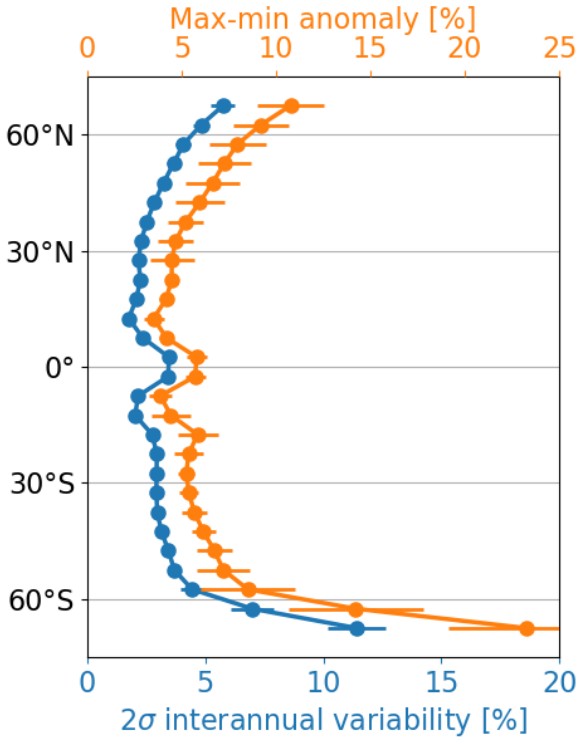

**Figure 4.** Interannual variability [%] of total ozone as a function of latitude derived from the GTO-ECV data record. The blue curve denotes two times the standard deviation of the yearly ozone anomalies based on the reference period 1997–2020, and the orange curve denotes the corresponding range (maximum-minimum) of the anomaly. The error bars denote the variation of the variability with longitude.

positive trend which we identified in the previous study for the tropics and which was attributed mainly to the impact of changes in ENSO intensity, is now insignificant for the entire region (except for the southern subtropics 20°–30°S). On the other hand, in the middle latitudes of the Southern Hemisphere we now find significant positive trends. In the middle latitudes of the Northern Hemisphere the spatial pattern remains more or less the same. Significant positive trends over the North Atlantic region and (non-significant) negative trends elsewhere were already found in the predecessor study.

In the Southern Hemisphere extratropical region it seems that the expected positive ozone trend related to decreasing amounts of ODSs becomes more and more evident. The trend does not indicate a significant longitudinal structure as found in the Northern Hemisphere, or an apparent correlation with the AAO proxy. On the other hand in the Northern Hemisphere it seems that the expected ODS-related trend is still masked by year-to-year variability (Weber et al., 2018; Braesicke et al., 2018; SPARC/IO3C/GAW, 2019). For illustration Fig. 4 shows the interannual variability of total ozone derived from the GTO-ECV data record during the period 1997–2020. As a measure for interannual variability (blue curve) we take two times the standard deviation of the yearly ozone anomalies based on the reference period 1997–2020. The error bars denote the variation of the variability with longitude. The variability is around 2% in the subtropics. Toward the middle and high latitudes it increases

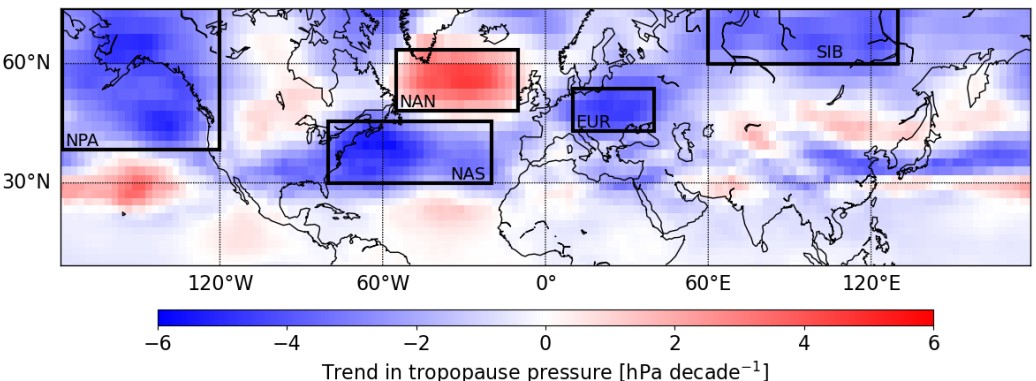

**Figure 5.** Linear trend 1997–2020 in tropopause pressure [hPa decade$^{-1}$] derived from the NCEP/NCAR reanalysis data. Black rectangles denote five regions for which we provide the mean and maximum trend in Table 3. These regions are the northern Pacific (NPA), the northern part of the North Atlantic (NAN), the southern part of the North Atlantic (NAS), Europe (EUR), and Siberia (SIB).

to ∼5% (see also Braesicke et al. (2018)). It is maximum (up to 10%) in the Southern Hemisphere, mainly determined by the year-to-year variation of the Antarctic ozone hole. There is a small local maximum in the inner Tropics related to the

regular QBO variations. The orange curve denotes the range (maximum minus minimum) of the yearly anomalies, which is between 10% in the Northern Hemisphere (60°–70°N), 5% in the Tropics, and 20% in the Southern Hemisphere (60°–70°S). The numbers retrieved for the interannual variability of total ozone underpin the challenge regarding the detection of expected ozone trends of the order of ∼1–2% relative to the year-to-year variation of ∼5% in the middle latitudes.

As mentioned earlier, the spatial pattern of the ozone trend in the middle latitudes of the Northern Hemisphere indicates a

correlation with the AO proxy. In order to elaborate this in more detail, we apply a simple linear regression to a tropopause altitude data product for the period 1997–2020. On a global scale, the trend in tropopause altitude is positive over recent decades and primarily thermally driven due to warming of the troposphere as a consequence of increasing amounts of greenhouse gases (Santer et al., 2003a, b; Pisoft et al., 2021). However, some notable latitudinal and longitudinal variation in the tropopause altitude trend as well as upward/downward trend dipoles were found (Xian and Homeyer, 2019). Since we do not see an ap-

parent correlation between the spatial pattern of the ozone trend and the AAO proxy in the Southern Hemisphere, we limit our analysis to the Northern Hemisphere (0–70°N). We use the NCEP (National Centers for Environmental Prediction) reanalysis derived tropopause altitude data product (i.e., monthly mean values on a regular grid of 2.5°×2.5°in latitude and longitude) that is provided by the National Oceanic and Atmospheric Administration/ Oceanic and Atmospheric Research/ Earth System Research Laboratories (NOAA/OAR/ESRL) Physical Science Laboratory (PSL), Boulder, Colorado, USA (Kalnay et al.,

270   1996).





**Table 3.** Average and maximum linear trend in tropopause pressure in hPa decade$^{-1}$ as well as average linear trend in total ozone and corresponding $2\sigma$ uncertainty (in parenthesis) in % decade$^{-1}$ for five selected regions in the Northern Hemisphere (see also Fig.5, black rectangles).

| Region | Abbreviation in Fig. 5 | Latitude | Longitude | Tropopause mean trend [hPa decade$^{-1}$] | Tropopause maximum trend [hPa decade$^{-1}$] | Total ozone mean trend [% decade$^{-1}$] |
|---|---|---|---|---|---|---|
| Northern Pacific | NPA | 40°–70°N | 180°–120°W | -3.0 | -5.2 | -0.4 (1.0) |
| North Atlantic northern part | NAN | 50°–62.5°N | 55°–10°W | 2.0 | 4.6 | 1.2 (1.2) |
| North Atlantic southern part | NAS | 30°–47.5°N | 80°–20°W | -3.2 | -5.6 | -0.4 (0.8) |
| Europe | EUR | 45°–55°N | 10°–40°E | -3.5 | -4.6 | -0.6 (0.8) |
| Siberia | SIB | 60°–70°N | 60°–130°E | -3.0 | -4.2 | -0.1 (1.5) |

The estimated linear trend for the tropopause pressure is shown in Fig. 5. In addition, Table 3 denotes the mean and the maximum values of the trend in tropopause pressure for five selected regions (northern Pacific, North Atlantic northern and southern part, respectively, Europe, and Siberia), which are marked by the black rectangles in Fig.5. In addition, the last column of Table 3 denotes the corresponding mean trend in total ozone. For the northern Pacific, for both the southern and northern part of the North Atlantic, as well as for Europe we find positive correlations with the trend in ozone. The regions indicating a negative trend in ozone (see Fig.3) also show a negative trend in tropopause pressure between -3.0 – -3.5 hPa decade$^{-1}$. The negative pressure trend corresponds to an increase of about 100–130 m decade$^{-1}$ in tropopause height. On the other hand, the northern part of the North Atlantic (NAN) indicates both a positive trend in ozone ($1.2\pm1.2\%$ decade$^{-1}$) and in tropopause pressure, respectively. The latter is about 2 hPa decade$^{-1}$ in that region.

The positive correlation between tropopause pressure and total ozone has been described in, e.g., Hoinka et al. (1996), Steinbrecht et al. (1998) or Varotsos et al. (2004). Short-term as well as long-term variations in ozone have been associated with changes in tropopause height. For the European station Hohenpeißenberg it was found that about 25% of the negative long-term trend in total ozone could be attributed to a tropopause that moved up by $150\pm70$ m decade$^{-1}$ (Steinbrecht et al., 1998). However, longitudinal and latitudinal variation of the relation between both variables was observed (Varotsos et al., 2004). Because of the apparent correlation we conclude that the spatial pattern in the ozone trend (Fig. 3) is at least partly induced by changes in tropopause altitude that vary with latitude and longitude. However, over Siberia the correlation between the trend in tropopause and ozone is less clear. The trend in total ozone is very close to zero ($-0.1\pm1.5\%$ decade$^{-1}$), whereas the change in tropopause pressure is about -3.0 hPa decade$^{-1}$.

## 4.2 Seasonal dependence of trends

The seasonal dependence of the total ozone trend obtained from the gridded GTO-ECV data record is shown in Fig. 6. It has been determined by expanding the fit coefficient for the trend as described in Eq. 2. In general, the spatial pattern of the trend



**Figure 6.** Total ozone trend 1997–2020 in % decade$^{-1}$ derived from the GTO-ECV data record as a function of season: (a) December, January, February (DJF), (b) March, April, May (MAM), (c) June, July, August (JJA), and (d) September, October, November (SON). Grey dots denote that the trend is statistically not significant.

in each season is very similar to the pattern of the annual mean trend (see Fig. 3), and latitudinal and longitudinal variations do not change significantly in course of the year.

However, small seasonal changes are found in the Northern Hemisphere in the North Atlantic region and over eastern Europe. In the North Atlantic region between 45° and 70° N we note that the significant positive trend is most pronounced between December and February. This is in agreement with Szeląg et al. (2020, their Fig. 4), they analyzed zonal mean trends and found positive trends mainly in the upper stratosphere with a maximum in local winter. Additionally, our results using the gridded dataset reveal that over the eastern part of Europe the trend is negative and significant for a small region between 40°–50° N and 30°–50° E (see also Fig. 3). Contrary to the positive trend over the North Atlantic, the negative trend is most





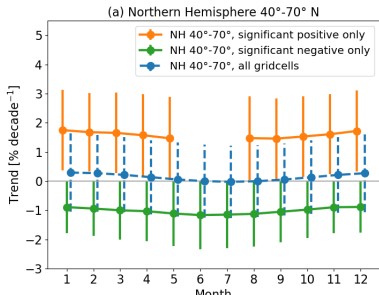
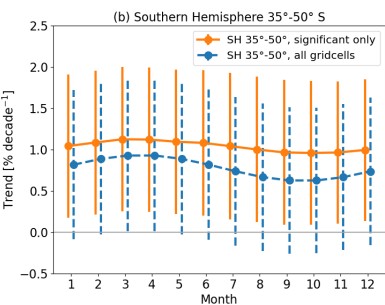
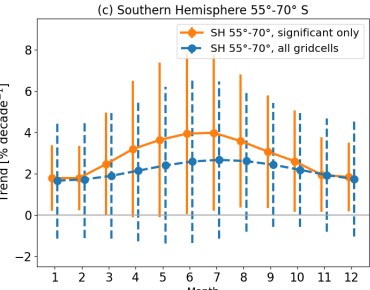

**Figure 7.** Total ozone trend as a function of month for three different latitude bands: (a) Northern Hemisphere $40°-70°$ N, (b) Southern Hemisphere $35°-50°$ S, and (c) Southern Hemisphere $55°-70°$ S.

pronounced from June to August (Fig. 6(c)). Moreover, this seasonal pattern agrees with the general conclusion of Bozhkova et al. (2019), who analyzed ozone trends in the middle latitudes of the Northern Hemisphere and found an overall increase in winter and a continuing decrease in summer for the period 2000–2017. In the previous section we described the correlation of the spatial pattern of the annual mean ozone trend in the Northern Hemisphere with the linear trend in tropopause pressure. Please note that for the trend in tropopause pressure we could not identify a corresponding distinct seasonal pattern.

In the tropical region ($30°$N–$20°$S) total ozone trends are not significant throughout the whole year. This is probably caused by the quite variable and ambiguous seasonal and altitude dependence of the ozone trends in this region (Szeląg et al., 2020). The equatorial band $10°$S–$10°$N indicates non-significant negative trends which are most pronounced from March to May, and which are possibly governed by seasonal variation in the lower stratosphere.

     In the extratropical region of the Southern Hemisphere significant positive trends are found throughout the whole year.
However, in the middle latitudes ($35°-50°$ S) the positive trends found in the Pacific, south of southern Africa, and south of Australia are strongest between March and May. Further south, in the latitude range $55°-70°$ S the positive trend is most pronounced in austral winter, i.e. between June and August (cf. Szeląg et al. (2020), their Fig. 4). This seasonal pattern observed in the high latitudes could be related to meridional transport processes that are controlled by the BDC. This is most active in local winter (Butchart et al., 2006) and a strengthening and acceleration of the BDC is prevailing in these months as well (Li
et al., 2008).

     For a more detailed illustration of the seasonal dependence we investigate the trend in three latitude bands $40°-70°$N, $35°-50°$S, and $55°-70°$S, respectively, for which we found significant trends in total ozone columns at least for some regions. Figure 7 denotes the trend as a function of month for the three selected zonal bands. The trend is shown averaged over all grid cells in this band (dashed blue curves), averaged over all grid cells which indicate significant positive trends (solid orange
curves), and, in case of the Northern Hemisphere averaged over all grid cells indicating a significant negative trend (solid green curve in Fig.7 (a)).

     In the middle latitudes of the Northern Hemisphere (Fig.7(a)) the positive trend is maximum in boreal winter from December to March, although the seasonal variation is very small ($\pm 0.1\%$) and not significant. The average positive trend (solid orange





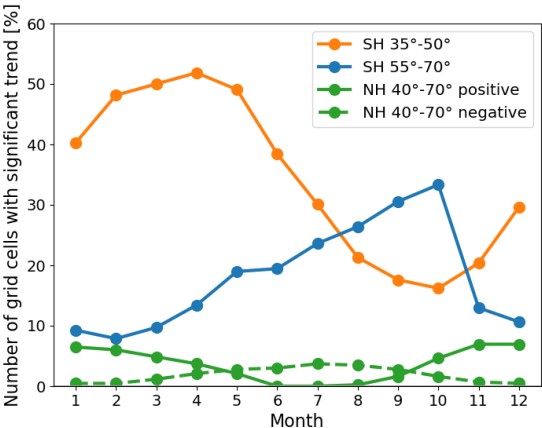

**Figure 8.** Number of grid cells indicating a significant trend as a function of month for the latitude bands $40°$–$70°$N (green), $35°$–$50°$S (orange), and $55°$–$70°$S (blue). The number is provided in percent of the total number of grid cells in the respective band. For the Northern Hemisphere (green) a division between significant positive (solid curve) and significant negative trends (dashed curve) is made.

line), which comes mostly from grid cells in the North Atlantic region, is $\sim$1.7% decade$^{-1}$, and the mean negative trend (solid

green line, mostly from grid cells in the eastern part of Europe) is about -1.1% decade$^{-1}$. If the estimated trends from all grid cells in this band are averaged (dashed blue curve), the trend is close to zero and not significant, which is in line with studies analyzing zonal mean ozone data records, e.g., Weber et al. (2018).

In the middle latitudes of the Southern Hemisphere ($35°$–$50°$S, Fig. 7(b)) the trend is $\sim$1.0% decade$^{-1}$ in those regions indicating a significant positive trend, and the maximum occurs between March and May. During these months the corresponding

zonal mean trend (all grid cells) is also on the edge of being significant (dashed blue curve). Figure 7(c) shows the seasonal variation of the trend in the band $55°$–$70°$S. The trend itself and the seasonal variation is much larger compared to the other bands. For the grid cells indicating a significant positive trend it varies from +2% decade$^{-1}$ (December to February, local summer) to +4.0% decade$^{-1}$ in local winter from June to August. Although the trend is found to be maximum in July, the clarity of the significance of the positive trend in this zonal band is higher in August ($3.6\pm3.2$% decade$^{-1}$) and September ($3.1\pm2.7$%

decade$^{-1}$). Significant positive trends in September and a strong seasonality were also found in Antarctica ($63°$–$90°$S) by, e.g., Solomon et al. (2016, 2017) or Kuttippurath and Nair (2017). Contrary to our results these studies indicate a maximum trend between September and December, whereas the minimum occurs from May to July. Note that our results cover only the latitude band $55°$-$70°$S and are not representative for the entire polar cap as the aforementioned studies. Moreover, this latitude band is significantly affected by the limited coverage of the satellite data during local winter from June to August and the results in

these months should be interpreted with great care.

To further illustrate and underpin the seasonal variation of the trend, Fig. 8 denotes the percentage of grid cells which indicate significant trends in the respective latitude band. The zonal bands and time periods showing the largest percentage of significant positive trends in total ozone are March and April in the middle latitudes of the Southern Hemisphere (50–55%) and September



to October in the band 55°–70°S (∼30%). The percentage of grid cells indicating significant trends in the Northern Hemisphere

(green curves) is much smaller compared to the Southern Hemisphere, except for positive trends from December to February, when about 15% of the grid cells indicate a positive trend. For the negative trend in the Northern Hemisphere the maximum percentage occurs from June to August, but it is well below 10%. As seen in Figs. 3 and 6 the regions showing significant trends in the Northern Hemisphere are limited to the North Atlantic sector and the southeastern part of Europe.

## 5 Summary

In this work we present an updated and detailed perspective on near-global (70°N–70°S) total ozone trends for the last quarter century. We use the GTO-ECV data record which has been extended and generated in the framework of the ESA-CCI and EU-C3S ozone projects. GTO-ECV combines total ozone observations from six nadir-viewing satellite sensors of the GOME-type. The latest addition was data from the TROPOMI instrument launched in October 2017. GTO-ECV covers the period from July 1995 through December 2020 and provides an excellent spatial resolution of $1° \times 1°$.

The main key aspects of our study is the analysis of latitudinally and longitudinally resolved patterns of the ozone trend as well as the investigation of its seasonal dependence. We evaluate the linear trend using a standard MLR method including several explanatory variables such as the Quasi-Biennial Oscillation, the solar cycle, El Niño–Southern Oscillation and Arctic/Antarctic Oscillation. For trend estimation we average GTO-ECV data on a somewhat larger grid size of $5° \times 5°$ and apply the approach to each latitude-longitude bin separately.

In the Southern Hemisphere we found statistically significant positive trends that increase from $0.6\pm0.5\%$ decade$^{-1}$ in the subtropics to $1.0\pm0.9\%$ decade$^{-1}$ in the middle latitudes and $2.8\pm2.6\%$ decade$^{-1}$ in the latitude band 60°–70°S. In the middle latitudes of the Northern Hemisphere the trend exhibits distinct regional patterns, i.e. latitudinal and longitudinal structures. Significant positive trends ($\sim1.5\pm1.0\%$ decade$^{-1}$) over the North Atlantic region as well as negative trends (-$1.0\pm1.0\%$ decade$^{-1}$) over Eastern Europe were found. The northern Pacific indicates non-significant negative trends. Furthermore, these

trends exhibit a correlation with long-term changes in tropopause pressure (see also Zhang et al. (2019)). The hemispheric differences indicate that in particular in the Northern Hemisphere the ozone trend is determined by both dynamic as well as ODS-related effects, which may induce changes of opposite signs. Total ozone trends in the tropics are not significant, which is most likely a result of opposed trends at different altitudes in the troposphere and stratosphere (Meul et al., 2016; Szeląg et al., 2020).

Regarding the seasonal dependence of the trends we found only very small variations over the course of the year. However, we identified different behavior depending on latitude. In the latitude band 40°–70°N the positive trend maximizes in boreal winter from December to February. In the middle latitudes of the Southern Hemisphere (35°–50°S) the trend is maximum from March to May. Further south toward the high latitudes (55°–70°S) the trend denotes a relatively strong seasonal cycle which varies from $\sim2\%$ decade$^{-1}$ in December and January to $\sim3.8\%$ decade$^{-1}$ in June and July (austral winter). The clarity

of the significance in this band is maximum in August and September. The maxima in local winters in both hemispheres might be related to the impact of the meridional transport of ozone (and other trace gases) that is controlled by the Brewer-Dobson

circulation. An acceleration due to increasing amounts of greenhouse gases is expected which might induce the observed seasonal behavior, but also contribute to long-term changes in ozone in addition to ODSs-related increases (Harris et al., 2008; Weber et al., 2021a).

Our study reveals that continued monitoring of the ozone layer and further extension of high-quality ozone data records, that provide reliable information about the temporal evolution and spatial variability, is needed. This will enable us to improve our current understanding of past changes in ozone and to gain more confidence in the attribution of the trends to various processes.

*Data availability.*   The GTO-ECV data record is publicly available via the Copernicus Climate Data Store (https://cds.climate.copernicus.eu). The level-2 ozone products from GOME, SCIAMACHY, OMI and GOME-2 are available on the Ozone_CCI ftp site at the following address:
ftp://cci_web@ftp-ae.oma.be/esacci. The TROPOMI GODFIT ozone products are publicly available on the Copernicus Sentinel-5P data hub (https://s5phub.copernicus.eu, DLR/BIRA-IASB/ESA/EU, 2021).

## Appendix A: Explanatory variables

Figure A1 shows the time series 1997–2020 of all explanatory variables used in the MLR (Eq. 1) normalized to zero-mean and unit-variance. Black curves in (e) and (f) denote a 3 month running mean.

*Author contributions.*   MCE and DL were responsible for the generation of the GTO-ECV data record. MCE performed the trend analysis, and CL and MVR were responsible for the generation of Level-2 ozone data records. All authors contributed to the revision of the manuscript.

*Competing interests.*   The authors declare that they have no conflict of interest.

*Acknowledgements.*   NCEP Reanalysis Derived data were provided by the NOAA/OAR/ESRL PSL, Boulder, Colorado, USA. We thank EU/ESA/DLR/BIRA-IASB for providing the TROPOMI/S5P Level-2 ozone products, this paper contains modified Copernicus data (2018/2020)
processed by DLR.

*Financial support.*   MCE and DL acknowledge support from the DLR projects MABAK and INPULS.



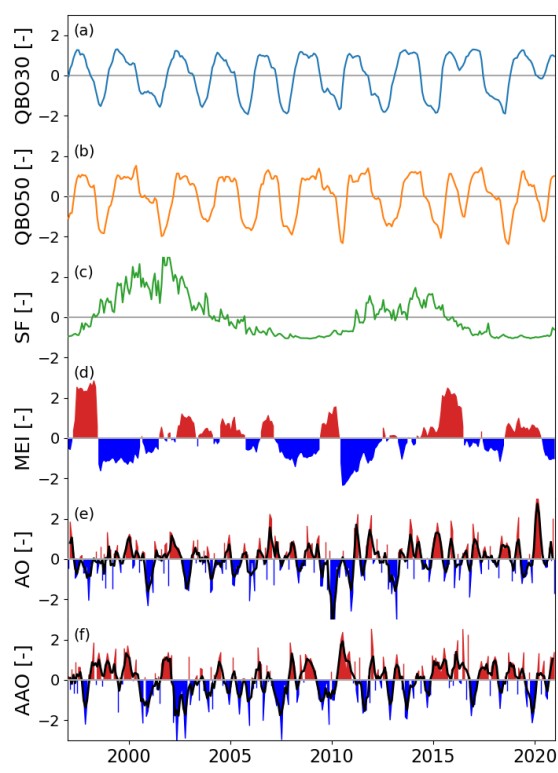

**Figure A1.** Time series 1997–2020 of all explanatory variables used in the MLR (Eq.1) normalized to zero-mean and unit-variance: (a) QBO30, (b) QBO50, (c) SF, (d) MEI, (e) AO, and (f) AAO. Black curves in (e) and (f) denote a 3 month running mean.

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
