# Peer review of "Global, regional and seasonal analysis of total ozone trends derived from the 1995–2020 GTO-ECV climate data record"

_Atmospheric Chemistry and Physics, 2021_

## Author Comment (AC1)

*Response to reviewer #1*

*We thank reviewer #1 for her/his valuable comments. Please find below the reviewer's comments (in black), our responses (in blue), and changes or additions to the text (in red). All page / line numbers refer to the old version of the manuscript.*

This is the first review of the paper "Global, regional and seasonal analysis of total ozone trends derived from the 1995–2020 GTO-ECV climate data record" by Melanie Coldewey-Egbers et al.

This paper is focused on updates of the trends derived from the extended GTO-ECV satellite total ozone record. The dataset was recently updated by including new satellite records (i.e. TROPOMI) and by extending previously used satellite records. The dataset is homogenized by using the same processing algorithm for all satellite records.

In the paper, the authors apply a statistical model to the GTO-ECV record to detect ozone changes over a wide range of latitudes (70S-70N). The regional and seasonal patterns in both Hemispheres are identified and linked to the chemical and dynamical processes that contribute to ozone recovery.

This paper is an extension to the previously published analyses by Coldewey-Egbers et al. (2014), however in this paper statistical model also includes additional explanatory parameters (AO and AAO), which improve the model fit to the data, especially at high latitudes, thus suggesting a significant contribution of the changes in the Brewer-Dobson circulation to the apparent ozone recovery in some regions.

Among new findings, authors report that positive trends in the 50-70 S region are highly significant in many regions. Since the AAO controls the transport of ozone to high latitudes, did you check if the AAO contribution to the ozone variability has significantly increased over the last 25 years?
→ We checked if the AAO contribution to ozone variability has changed using two options:
  1. Keep the start date of the regression at 1997 and vary the end date of the regression, i.e. vary the length of the time series
  2. Vary the start and end date of the regression using a fixed length of 16 years.
→ We did not find an increase in the AAO contribution and did not see a change in the spatial pattern.

This paper provides a strong contribution to the understanding of the drivers of ozone recovery.

The figures are clear and support the discussion and main conclusion points of the paper.

**Comments:**

Line 30, "(though not statistically significant)" I recall that Ball et al. 2018 claimed a statistically significant ozone decrease in the lower stratosphere. He stated that the models did not support this finding. Please clarify if you are discussing total stratospheric ozone or low stratospheric ozone changes. I believe Ball et all (2019) has reassessed the significance of observed and modeled changes in the datasets extended to 2018, but still found high probability (90 % in NH mid-latitudes, 95 % in tropics, and 80 % in the SH middle latitudes) that the lower stratospheric ozone declined by 2018 as compared to the 1998 levels. I think it might be good to state in this paragraph that analyses were done using broad latitude bands and zonal averages, while your paper has assessed zonally resolved data for trends.
→ We have deleted the statement "(though not statistically significant)" since this is not correct as pointed out by the reviewer. We have added the reference Ball et al., 2019 here and mention that the cited studies mainly use zonally averaged data in contrast to our study.

Line 35, "changes in the tropospheric amount do not indicate a clear global pattern" – this paper was published back in 2018. Should a more recent assessment of tropospheric ozone trends be used here (i.e. Ziemke et al. 2019) that found regional patterns in TOR trends?
→ We have replaced the reference Gaudel et al., 2018 with Ziemke et al., 2019, as suggested and adapted the text accordingly:
"On a global scale ozone in the troposphere has increased during the past decades. However, the enhancement is highly regional (Ziemke et al., 2019)."

Line 45, you mentioned "focusing on different questions". Please indicate how this paper's focus is different from other papers.
→ Since referee #2 raised a similar comment, we rephrased the sentence:
"… focusing on various questions such as the consistency of trends and their uncertainties derived from different data sets, as well as the dependence of the trends on latitude, altitude, or season."

Line 64 "inter-relation" vs correlation: please clarify what it means.
→ We meant "correlation".

Lines 122-124. Do you introduce seasonal variability in the trend term only or also in coefficients of other proxies?
→ We introduce seasonal variability only in the trend term, but not in the other explanatory variables. We now explicitly mention this in the text.

Line 167, you can also add Appenzeller et al, 2000 and Rieder et al (2010) papers to the references.
→ We added Rieder et al., 2010 in the discussion of the impact of the AO/NAO (line 176).

Line 175 "over Iceland" – the anomaly pattern in the figure is not centered on Iceland. Should the description be altered to describe the are (i.e. geographical coordinates of the region).
→ We altered this sentence to:
"The corresponding lower tropopause altitude leads to an increase in total ozone (i.e. a positive correlation between AO and ozone, which is centered over the Labrador Sea),..."

Lines 261-263. Have you considered the change in the position of the subtropical and polar jets that can contribute to regional changes in the total ozone (i.e. Seasonal and Regional Variations of Long-Term Changes in Upper-Tropospheric Jets from Reanalyses by Gloria L. Manney and Michaela I. Hegglin in J of Clim, 2018)?
→ Thank you very much for pointing to this study. We did not consider changes in the jets in our study.

Line 271-272, please explain what you mean by the "maximum values of the trends".
→ This is just the trend value for that grid cell which shows the maximum trend in the respective region.

Lines 286-289, Can the total ozone trend over Siberia be offset by an increase in tropospheric ozone due to wildfires? It might be interesting to investigate seasonal differences in the tropopause height trends.
→ An increase in tropospheric ozone can contribute to the trend. Unfortunately we could not find a study providing robust tropospheric ozone trend estimates for Siberia. However, according to Ponomarev et al. (2016) the number of wildfires over Siberia steadily increased over the past decades which supports your hypothesis.
→ Regarding the trend in tropopause height, we did not find a seasonal dependence. Ponomarev, Evgenii I., Viacheslav I. Kharuk, and Kenneth J. Ranson. "Wildfires Dynamics in Siberian Larch Forests" *Forests* 7, no. 6: 125. https://doi.org/10.3390/f7060125, 2016.

Figure 6. since the differences are hard to discern, would it make sense to make plots of the seasonal differences from the annual trends?
→ Thank you for this suggestion. We created these plots, but in our view they would lead to more confusion instead of clarity. Thus, we would prefer to leave Fig 6 as is.

---

## Author Comment (AC2)

*Response to reviewer #2*

*We thank reviewer #2 for her/his valuable comments. Please find below the reviewer's comments (in black), our responses (in blue), and changes or additions to the text (in red). All page / line numbers refer to the old version of the manuscript as published in ACPD.*

**General comments**

The manuscript "Global, regional and seasonal analysis of total ozone trends derived from the 1995–2020 GTO-ECV climate data record" by Melanie Coldewey-Egbers et al. analyses the regional patterns of the total column ozone 1997-2020 trends estimated by MLR on the updated GTO-ECV satellite data record. Regions presenting significant positive and negative trends are highlighted in the Northern and Southern Hemispheres. Ozone trend patterns are correlated with long-term changes in the tropopause pressure. Finally, the seasonal dependence of the regional trends is investigated. For each latitude band, the seasonal variation of the mean trend is compared to the seasonal variation of the significant regional trends. The seasonal variation of the trend is also illustrated by the variation of the percentage of cells of significant trends in each latitude band. The regional and seasonal variations are qualitatively linked to chemical and dynamical processes.

The paper is clear and well written with good quality figures. The scientific contribution is highly relevant for publication and fits the scope of ACP. The methods used are valid and support the claims made by the authors. The paper will make a very good contribution to ACP and I recommend its publication, provided that the following comments are addressed.

**Specific comments**

P2, L26: please add a reference for "the overall total ozone levels still remain lower than the pre-1980s values"
→ Braesicke et al. (2018) has been added.

P2, L34: "Another confounding factor contributing to the overall trend in the total column is the evolution in the troposphere in particular in the tropics", please rephrase like: Another confounding factor in determining the overall trend of the total column is the evolution in the troposphere, particularly in the tropics.
→ Sentence has been rephrased as suggested.

P2, L43: "In addition... other projects...various trends...on differents questions": this sentence is too general (other, various, differents), please be more specific.
→ We provide more details:
"In addition to ESA-CCI other projects and initiatives aimed at generating long-term data sets of total columns and profiles, e.g., the University of Bremen total column record (Weber et al., 2022), the National Aeronautics and Space Administration (NASA) Merged Ozone Data Set (MOD, Frith et al., 2014), or the Global OZone Chemistry And Related trace gas Data records for the Stratosphere (GOZCARDS, Froidevaux et al., 2015). All have recently been used for

trend analyses focusing on various questions such as the consistency of trends and their uncertainties derived from different data sets, as well as the dependence of the trends on latitude, altitude, or season (see, e.g., Weber et al., 2018; SPARC/IO3C/GAW, 2019; Szelag et al., 2020; Weber et al., 2022, for more details)."

P2, L51: "The main findings of Coldewey-Egbers et al. (2014) were positive, though non-significant, ozone trends for large parts of the globe (60° N–60°S), in particular in the middle latitudes": What is particular in the middle latitudes, positive or non significant trends? Fig1a of the Coldewey-Egbers et al. (2014) paper shows significant positive trends for a large part of the globe in 30°S-30°N. Is this also a main finding of the paper? As the significance is a main difference between the two studies, this should be mentioned.
→ We rephrased this part, so that the statement becomes clearer. In the middle latitudes the trends are mostly not significant. Additionally, we mention the significance in the tropics in CE2014.

P2, L58 and P3, table1: The difference between the two data records (Coldewey-Egbers et al. (2014) and this study) is not only the 3 additional sensors but also the used time range for GOME and SCIAMACHY. This is mentioned P3 L94 as "minor changes related to time periods". This should be mentioned and the reason should be given. Reference to Coldewey-Egbers et al. 2020 or Garane et al. 2018.
→ We added the following explanation on p.4:
"For GOME we have extended the time period in order to improve the representativeness of GTO-ECV between 2003 and end of 2004, and for SCIAMACHY and GOME-2A we had to shorten the periods based on updated comparisons with the reference sensor OMI and ground-based measurements (Garane et al., 2018; Coldewey-Egbers et al., 2020)."

P9, L75: While "ESA-CCI" is defined P2, L39, "ESA-CCI+" is not defined. Is it a simple extension?
→ Yes, ESA-CCI+ is the successor initiative. We now mention this explicitly in the text.

P4, L94: Removing 7 years from 2 over 6 data records is not a minor change. Please explain.
→ See also reply to comment "P2, L58". We now provide an explanation in the text.

P5, L137: Fig. A1 is not essential. Think to remove it.
→ We now refer to this Figure in two more places in the manuscript. Therefor, we would prefer to keep it.

P7, Fig1c: The significance of the SF explanatory variable is very different from Coldewey-Egbers et al. (2014) Fig1e. Could you give any reason/explanation?
→ Regarding the differences in the significance of the solar cycle proxy between the old and the new study, we assume that this could be related to the length of the time series. The old study covers only 1.5 cycles, whereas the new study covers slightly more than 2 complete

cycles. According to Steinbrecht et al. (2004) magnitude and timing of ozone variations in response to the solar cycle can be different from one cycle to the next.
[Steinbrecht, W., Claude, H., and Winkler, P., Enhanced upper stratospheric ozone: Sign of recovery or solar cycle effect? *J. Geophys. Res.*, 109, D02308, doi:10.1029/2003JD004284, 2004.]

P8, L193: "Note that we do not expect significant changes regarding the ozone trend when we include the additional terms in the MLR, but uncertainty should be further reduced" Have you tested that by estimating trends with and without the additional term?
→ We tested this and compared the trends with and without the additional term. We found that the spatial pattern and the magnitude remain roughly the same. However, small changes were found, and for a small number of grid cells the significance of the trend changed from "significant" to "not significant" and vice versa. We think that the changes might be induced by the impact of a possible long-term drift in the additional terms (AO/AAO, see also Weber et al., 2021a and Hu et al., 2018). We added a comment in the text.

P10 L205-209: The positive total column ozone trends patterns are compared to the upper stratospheric trends patterns in Sofieva et al 2021, and the negative trends patterns are compared to the middle stratospheric trends patterns of Sofieva et al, 2021. Could you give a reason why the total column ozone trend is related to upper stratosphere at one place and to middle stratosphere at another place?
→ We realized that our phrasing was inappropriate here. We reformulated this part (ll.206-209) to:
"Their height-resolved trends also indicate in the Northern Hemisphere a longitudinal dependence. Positive trends were found over Scandinavia and the North Atlantic at almost all altitudes. They are statistically significant in the upper stratosphere above 40km. Non-significant negative trends were found by Sofieva et al. (2021) below 40km in the same regions as found here..."

P10, L211: "..., which cannot be easily separated from the ODSs-related trends." Add: "..., considering the explanatory variables used in this study"
→ Done.

P10, L211: "...indicates some correlation with ...": actually, we see a positive correlation in NH but a negative or no correlation in SH. Explanations are needed and can be found later in the text. Please refer to the adequate section(s) here.
→ We rephrased this sentence, since the statement regarding the correlation should be limited to the Northern Hemisphere, and we refer to the end of the section for more details.
"The spatial trend pattern in the Northern Hemisphere described above indicates some positive correlation with the fit coefficient for the AO proxy (Fig.2(a))."

P10, L217: "Moreover, it is not significant from 30°N to 20°S" But this is not the case with the 2014 record. Do you have an explanation for this change?

→ In Coldewey-Egbers et al. (2014), we attributed the positive trends found in the tropics to the impact of ENSO. During the period 1995-2013, ENSO was dominated by a large positive event at the beginning of the period and a significant negative event at the end (2011-2012) (see also Fig. A1 in the recent paper), which may have induced the trend in ozone. On the other hand, the extended period (1995-2020) contains another large positive ENSO event in 2015, which may then be the reason for the observed differences in the ozone trend.

P10, L231: Is GTO-EVC included in the "other datasets" used in the Weber 2018 study? Could the differences in the used data record be a cause for the discrepancy mentioned in P10, L235?
→ Yes, GTO-ECV is used in both Weber et al. (2018) and Weber et al. (2022). We added this information here, and we reformulated this paragraph, since Weber et al. (2022) is now available in ACPD. We compare our results with the updated study, which does not show the mentioned discrepancy in the northern subtropics anymore.

P11, Fig4: Please consider adding a vertical grid to make the reading of the differences between NH and SH easier.
→ We added a vertical grid and adjusted the range of the second (orange) x-axis.

P11, L247-249: "On the other hand in the Northern Hemisphere it seems that the expected ODS-related trend is still masked by year-to-year variability (Weber et al., 2018; Braesicke et al., 2018;SPARC/IO3C/GAW, 2019). For illustration Fig. 4 ..." The authors say that it is more difficult to see the positive trend related to ODS in NH that in SH because the year to year variability is higher in the NH than in the SH. However, in Fig. 4, the blue (interannual variability) and the orange (yearly anomalies range) curves are not different in NH and in SH for the 30-60° latitude bands. The only difference is the higher error bars in NH, and these are indicating the variation with longitude. Do you mean that the higher variation with longitude in NH makes it so difficult to estimate the trend that it is more influenced by the AO than by the ODS decrease? If yes, I suggest you make this statement clearer in the paragraph by emphasing on the error bars values rather than on the variability values. If not, could you please explain what you mean?
→ The reviewer is right. The variability in the middle latitudes is the same for NH and SH. We removed the statement regarding the stronger variability in NH. We add a comment that the expected ODS-related trend in NH might be balanced by trends induced by dynamical factors (see also Weber et al., 2022).

P15,L304: "Please note that for the trend in tropopause pressure we could not identify a corresponding distinct seasonal pattern". How does this affect or change the conclusions that have been drawn from the correlation between the total column ozone trends and the tropopause height trends on P13 L284-286? Please comment in the text.
→ The long-term trend in tropopause pressure in the NH was mainly attributed to a warming of the troposphere as a consequence of increasing amounts of greenhouse gases (Santer et al. 2003a,b). We didn't find information regarding a possible seasonal dependence in the tropopause trend in the literature (except from a few regional studies), and a detailed analysis was beyond the scope of our study. Regarding the seasonal dependence of the ozone trend,

we conclude that other dynamical factors are responsible, e.g. changes in the wave-driven Brewer-Dobson circulation, whose impact maximizes during winter months (Butchart et al.,, 2006). Moreover, the strengthening and acceleration of the BDC is also maximum in winter (Li et al., 2008). We added a comment in the text just before the statement (p.15, l.304) cited by the reviewer.

P15, L308: "...which are possibly governed by seasonal variation in the lower stratosphere." Please reference this statement.
→ We added the reference "Szelag et al. (2020)" here.

P14, L 294: "small seasonal changes" to P15, L315: the description of the small seasonal changes is detailed and finally established as not significant when Fig 7 is described. Fig 7 is a nice representation of Fig 6 results facilitating the rather laborious visual comparison of the 4 panels by the reader, but, it shows clearly that the seasonal variation of the trend is not significant. This statement should be made explicit at the beginning of the seasonal variation description.
→ The end of the first paragraph in Sec. 4.2 now reads:
"..., and latitudinal and longitudinal structures do not change substantially in course of the year. On a global scale, clear and significant seasonal variations are not visible."
These changes are not significant in both cases (all cells in blue and significant trend cells in orange) as shown by the blue and orange error bars in Fig 7 and for all latitude bands. This should be explicitly repeated, not only for NH at P15 L323.
→ In the first paragraph discussing Fig.7 we added:
"The observed seasonal variation is not significant, neither for all grid cells in the latitude band (blue curves) nor for the grid cells indicating statistically significant trends (orange and green curves), as indicated by the corresponding error bars in Fig. 7."

P15, L316: "three latitude bands 40°–70°N, 35°–50°S, and 55°–70°S,..." These are not the classical latitude bands and have been chosen in function of the significant trend cells. However, the latitude bands results are compared with other studies considering the classical latitude bands (P16,L327). Would the statement "is in line with studies" (P16, L326) be the same if classical latitude bands would have been considered?
→ The statement "...is in line with studies..." refers to the latitude band 40°-70°N. We have repeated our analysis of the seasonal behavior using the 'classical' latitude band 35°-60°N (as for example used in Weber et al. (2018, 2022)), and the results are nearly identical. Thus, our statement regarding the agreement with Weber et al. (2018) would be the same.

P15, L318: "The trend is shown averaged over all grid cells in this band (dashed blue curves), averaged over all grid cells which indicate significant positive trends..." Please precise that the orange trends are not representative for the latitude bands but only for some regions of the bands.
→ We have added:
"Hence, the orange and the green curves represent the seasonal behavior of the trend only for some longitudes in the respective latitude band."

P16, L312: "Further south, in the latitude range 55°–70° S the positive trend is most pronounced in austral winter..." : The orange trends are not representative for the latitude bands but only for some regions of the bands. Here this statement is true only for some longitudes. Please rephrase.
→ We rephrased the sentence:
"Further south, in the latitude range 55°-70° S the positive trend, which is significant over the southern Pacific (110°-180° W) and around 0° E/W (see Fig.6), is most pronounced in austral winter..."

P16, L338: "Note that our results cover only the latitude band 55°-70°S..." please add: "...and consider only the longitudes regions where the trend is significant"
→ Done.

P16 Fig 8 and L341-348 : Fig 8 shows that the percentage of significant grid cells varies within the course of the year. This does not represent the seasonal variation of the trend value but the seasonal variation of the surface of significant trends. The first sentence of the paragraph may be confusing, please consider rephrasing.
→ We removed "of the trend" in this sentence.

P17, L365: "The hemispheric differences indicate that in particular in the Northern Hemisphere the ozone trend is determined by both dynamic as well as ODS-related effects, which may induce changes of opposite signs." Please complete with the second hemisphere: "while in the Southern Hemisphere ...."
→ We completed the sentence:
"..., while in the Southern Hemisphere the less pronounced longitudinal structures in the overall positive trends may suggest that ODS-related effects prevail, which is in good agreement with Weber et al. (2022)."

P17, L370: "only very small variations over the course of the year": please mention the non significance
→ Included.

P17, L375: "The maxima in local winters in both hemispheres": please mention the non significance
→ Included.

P17, L377: "An acceleration due to increasing...": replace with "..acceleration of the BDC due.." and add a reference please.
→ Done.

**Technical corrections**

P2, L32: add "s" to "Hemisphere"
→ Changed.

P2, L47: How can a 2015 paper (Coldewey-Egbers et al. (2015)) be the reference for a 2014 paper (Coldewey-Egbers et al. (2014))?
→ This reference has been deleted. It appears in Sec.2.

P3,L62: ...which is "a" key indicator of climate change...
→ Added "a"

P3, L71: evt replace "summary" by "conclusions"
→ Replaced.

P3, L73: remove "s" from "a series"
→ Maybe "a series" is correct? Not changed yet.

P3, L80: replace "ingested" by a less edible term like e.g. "included"
→ Changed to "incorporated".

P4, L97: "...an excellent long-term stability.": please add the reference for the validation of the updated data record: Garane et al 2018
→ Reference has been added.

P5, L122: move here "The sources and names of the selected explanatory variable time series are given in Table 2" of P5,L136
→ Done.

P5, L147: "...2021a), which might explain the lower values of the explained variance we achieve with the MLR." Please, rephrase like: "...2021a). This may explain the lower explained variance values..."
→ Sentence has been rephrased.

P7, L168: replace "The left panel of Fig.2" with "Fig 2a"
→ Done.

P8, L182: replace "Fig. 2" with "Fig. 2a"
→ Done.

P8, L189: replace "The right panel of Fig. 2" with "Fig. 2b"
→ Done.

P10, L211: replace "Fig. 2" with "Fig. 2a"
→ Done.

P10,L232: move "only in the middle latitudes of the Southern Hemisphere" after "were detected".
→ Changed.

P17, L345: please replace "except" with "but are similar"
→ Done.

P17, L346: "15%" : Fig 8 rather shows 10%.
→ Changed.

P16, Fig 8:
y label: the unit of a number of cells should not be given in %. Please adapt the label.
In the legend, the "NH 40°-70° negative" should be a dashed line.
→ Label has been adapted and a dashed line appears in the legend.

P17, L53: replace "data from" with "data record of"
→ Done.

P17, L355: remove "s" from "aspects"
→ Done.

P17, L365: "(see also Zhang et al. (2019))" the first occurrence of this reference is here in the conclusions. Please, place this reference before section 5.
→ We added the reference in Sec. 4.1.

P17, L366: remove "in particular"
→ Done.

P24, L574: the http address for (United Nations Environment Programme, 1986) points to a "page not found"
→ Replaced with the latest edition 2020: https://ozone.unep.org/sites/default/files/Handbooks/MP-Handbook-2020-English.pdf

---

## Author Comment (AC3)

*Response to reviewer #3*

*We thank reviewer #3 for her/his valuable comments. Please find below the reviewer's comments (in black), our responses (in blue), and changes or additions to the text (in red). All page / line numbers refer to the old version of the manuscript.*

**GENERAL**

The paper is dedicated to the updated GTO-ECV total ozone climate data record, and to evaluation of global, regional and seasonal ozone trends.

The paper is well-organized and written, and it contains important information. Please find my minor comments below.

**MAIN COMMENTS**

For reporting trends with uncertainty interval (x +-y%), please make clear that uncertainties are 2-sigma (I believe, they are 2-sigma). It can be done with the first such trend reporting.
→ We added the information in the abstract and in Sec. 4.1.

In addition to Figure 1, I would suggest adding a figure (maybe in appendix) showing the percent of variability explained by each proxy. For such analysis, 2 QBO components can be combined into one source. Such figure would be useful in visualization of relative contribution of each proxy to observed ozone variability.
→ As suggested, we added this Figure in the appendix. We provide the contributions for QBO30, QBO50, Solar Flux, and MEI in Fig. A2 (a)-(d), respectively. In order to be consistent with Figure 1, we show the contributions for QBO30 and QBO50 separately. We refer to this Figure in Sec. 3.
The contribution of the AO/AAO proxy is already indicated in Figure 2(b), which shows the difference between the MLR without and with AO/AAO. Therefore, we do not show this contribution in Fig. A2 again.

The analyses presented in the paper show pronounced dependence of ozone trends on tropopause altitude. The tropopause altitude can be also used as a proxy in regression. It would be interesting to assess the influence of using tropopause height as a proxy on estimated ozone trends.
→ As suggested, we used the tropopause height as a proxy in the regression (replacing the AO/AAO proxy). The spatial trend pattern did not change, but the trend estimates in particular in the Northern Hemisphere became smaller and remained non-significant. The areas, for which the trend was significant (North Atlantic and SE Europe), became also slightly smaller. The fit coefficient for tropopause pressure is positive for the entire globe and about 0.5DU/hPa in the middle latitudes of the Northern Hemisphere, which corresponds to about -15DU/km (at tropopause altitude). This is in good agreement with Steinbrecht et al. (1998), who assumed a value of about -16DU/km.

**DETAILED COMMENTS**

L. 30-31, Ball et al. (2018) reported statistical significance of the lower-stratospheric trends. It is worth to mention also recent studies - (Ball et al., 2019, 2020; Orbe et al., 2020; Dietmuller et al., 2021).
→ Additional studies have been added.

L.65 "inter-relation" – do you mean correlation?
→ Yes, we meant correlation. Changed.

L.88, 97: Please provide quantitative measures of "a very good quality", "very good overall agreement", "" excellent long-term stability".
→ We now provide numbers for all statements:

L.88 → "...evidences a very good quality. The mean bias between the individual level-2 products and ground-based measurements is within 1.5±1.0% (Garane et al., 2018, 2019)."

L.89 → "...the inter-sensor consistency of the selected instruments is overall extremely high (within 1.0% between 50°N and 50°S (Garane et al., 2018))."

L.97 → "...reveals a very good overall agreement (i.e., similar to the validation of the level-2 data and with 0.5% to 1.5% peak-to-peak amplitude) and an excellent long-term stability (Garane et al., 2018). The drift with respect to ground-based data is well below 1% decade$^{-1}$."

L. 100- 103: This sentence suits better for the discussion section.
→ We moved this sentence to the end of the discussion section.

L. 205-209: The comparison with height-resolved trends at different altitudes looks strange. In (Arosio et al., 2019) and (Sofieva et al., 2021), the longitudinal difference in trends between Scandinavia and Siberia are observed at all altitude levels in the stratosphere.
→ Reviewer #2 raised a similar comment and we rephrased this part.

L. 237 – this paragraph. Compared to Coldewey-Egbers et al. (2014), this paper uses not only the updated dataset, but also a different MLR. This is worth to note.
→ We have added this information in L238.

L.265-270. Reanalyses data can be not optimal for trend analyses, since changing number of assimilated datasets with time can introduce artificial steps in data (for example, Simmons et al., 2014). Is it checked that the NCEP trends in tropopause height are in good agreement with those from experimental data?
→ Using the NCEP/NCAR tropopause height was motivated by two studies (Santer et al., 2003a, 2003b), who used this reanalysis data set for the same purpose, i.e. the derivation of trends. Santer et al. (2003a) found a good agreement with changes obtained from radiosonde data, and the agreement with other reanalysis data set was also found to be reasonable.

Santer, B.D., R. Sausen, T.M.L. Wigley, J.S. Boyle, K. AchutaRao, C. Doutriaux, J.E. Hansen, G.A. Meehl, E. Roeckner, R. Ruedy, G. Schmidt, and K.E. Taylor: Behavior of tropopause

height and atmospheric temperature in models, reanalyses, and observations: Decadal changes. *J. Geophys. Res.*, **108**, no. D1, 4002, doi:10.1029/2002JD002258, 2003a

Santer, B. D., Wehner, M. F., Wigley, T. M. L., Sausen, R., Meehl, G. A., Taylor, K. E., Ammann, C., Arblaster, J., Washington, W. M., Boyle, J. S., & Brüggemann, W.: Contributions of Anthropogenic and Natural Forcing to Recent Tropopause Height Changes. *Science*, *301*(5632), 479–483. http://www.jstor.org/stable/3834678, 2003b

L.291. How many terms were used for characterization of seasonal dependence?
→ We use only one term ($N_b$=2) to account for annual variations.

I think that Figure 8 does not bring essential information – this is the summary of previous figures. Furthermore, the Hemispheres cannot be directly compared, as the latitude bands are different. I suggest placing this figure in Appendix.
→ Maybe the label/caption of Fig. 8 was a bit misleading. Reviewer #2 also points this out. We do not show absolute numbers of grid cells indicating a significant trend, but the percentage of significant grid cells in the respective latitude band. We changed the label/caption accordingly. We think that the three latitude bands can be compared and we would prefer to keep this Figure in Sec. 4.2.

L. 358-359. The last sentence of this paragraph contains technical information and can be omitted in Summary.
→ We have deleted this sentence.

L. 367 Please add "statistically" before "significant"
→ Done.

REFERENCES

Ball, W. T., Alsing, J., Staehelin, J., Davis, S. M., Froidevaux, L., and Peter, T.: Stratospheric ozone trends for 1985–2018: sensitivity to recent large variability, Atmos. Chem. Phys., 19, 12731–12748, https://doi.org/10.5194/acp-19-12731-2019.

Ball, W. T., Chiodo, G., Abalos, M., Alsing, J., and Stenke, A.: Inconsistencies between chemistry–climate models and observed lower stratospheric ozone trends since 1998, Atmos. Chem. Phys., 20, 9737–9752, https://doi.org/10.5194/acp-20-9737-2020, 2020.

Dietmüller, S., Garny, H., Eichinger, R., and Ball, W. T.: Analysis of recent lower-stratospheric ozone trends in chemistry climate models, Atmos. Chem. Phys., 21, 6811–6837, https://doi.org/10.5194/acp-21-6811-2021, 2021.
.
Orbe, C., Wargan, K., Pawson, S., and Oman, L. D.: Mechanisms linked to recent ozonedecreases in the northern hemisphere lower stratosphere, J. Geophys. Res., 125,https://doi.org/10.1029/2019jd031631, 2020.

Simmons, A. J., Poli, P., Dee, D. P., Berrisford, P., Hersbach, H., Kobayashi, S., and Peubey, C.: Estimating low-frequency variability and trends in atmospheric temperature using ERAInterim, Quart. J. Roy. Meteor. Soc., 140, 329–353, https://doi.org/10.1002/qj.2317, 2014.

---

## Author Response (AR2)

***Response to the Editor:***

*Dear Editor,*

*Thank you very much for your feedback. We considered all comments. Please find below the comments (in black) and our responses (in blue), and changes or additions to the text (in red).*

Comments to the author:
Dear Authors:
Thank you for revising the MS.
I find the revision OK and the MS is now ready to be published.
However, I have some suggestions here. Please consider them.
* * *
L1: Delete "In this study" and start "We present"
Done.

L4-5: to examine the regional patterns and seasonal dependency
Changed.

L8-9: the uncertainty and trend value are same (i.e. 1.0). Then how can it be significant? Please rewrite.
Rewritten to "barely significant negative trends".

L10: trends correlate with tropopause pressure?
Changed.

L31: Ball et al., 2018; 2019; 2020; need not write separately
Changed.

L32: increase in what? Ozone?
"in ozone" has been added.

L83: "and" Garane et al.
Changed.

L91: Total Column Ozone (TCO)
Changed.

L94: 2018; 2019
According to the Copernicus style file, it seems that papers from the same authors are separated by a comma. Kept as is.

L137: "The year 1997 is…. "
Changed.

L137: ODS peaked, where? Need to write the region.
"in the middle latitudes" has been added here.

L166: where are these upwelling and downwelling processes?
Explanation has been added.

L167: impact on ozone
Added.

L185: how much is this strong correlation, give a number
The correlation coefficient is ~0.3-0.4.

L219: between 50 "and" 60
Changed.

L263: "more evident" is enough, not more and more
Removed.

Table 3: Why there is a positive trend in northern north Atlantic?
The positive trend could be related to decreasing temperatures observed in this region (→ North Atlantic warming hole).

L318: this "between" is for latitudes or latitude and longitudes?
It is for latitudes and longitudes.

L325-326: Note that there is no corresponding seasonal pattern in the tropopause pressure. Rewrite THE sentence like this. Delete "Please".
Done.

L360-364: In addition to the points mentioned here, the studies also used high resolution ozone sonde data. This could be also a reason for the difference with the trends estimated from satellite data.
Done.

L379: "The key aspect"
Changed.

L404: "study shows"
Changed.